# Current Application of Mineralocorticoid Antagonist (MRA) in Heart Failure and CKD: Does Non-Steroidal Drug Add Novel Insights

**DOI:** 10.3390/biomedicines13071693

**Published:** 2025-07-10

**Authors:** Irene Carlino, Filippo Pirrotta, Luigi Gennari, Alberto Palazzuoli

**Affiliations:** 1Internal Medicine Unit, Department of Medical and Surgical Sciences, University of Siena, 53100 Siena, Italy; pirrottaf90@gmail.com (F.P.); luigi.gennari@unisi.it (L.G.); 2Cardiovascular Diseases Unit, Cardio Thoracic and Vascular Department, Le Scotte Hospital Siena, 53100 Siena, Italy

**Keywords:** heart failure, chronic kidney disease, MRAs, finerenone, type 2 diabetes

## Abstract

Heart failure (HF) treatment evolved in the last 5 years with the introduction of new agents capable of reducing HF hospitalization and HF-related mortality. However, some categories such as patients with renal dysfunction tend to be excluded from larger randomized clinical trials. Additionally, most patients with HF experienced unavoidable glomerular filtration rate (GFR) deterioration during the clinical course. This is related to both cardio–renal interaction pathways and common cardiovascular risk factors that affect HF and chronic kidney disease (CKD). However, mineralocorticoid antagonists (MRAs) remain a cornerstone of HF therapy regardless of left ventricular ejection fraction (LVEF) values; some concerns remain about their utilization in CKD. Nevertheless, three studies (FIDELIO, FIGARO, and FINEARTS) have recently showed beneficial effects in both patients with HF and CKD associated with diabetes. Notably, finerenone a new non-steroidal MRA represents a significant step forward in cardiovascular therapy; its application spans a wide spectrum of HF phenotypes and CKD stages, and ongoing investigations will further elucidate its role in combination regimens and in broader patient populations. Further study may investigate the role of the drug in patients with heart failure with reduced ejection fraction (HFrEF) and in the severe CKD stage of non-diabetic etiology. In the current review paper, we provide a chronological overview of major trials evaluating the renal outcomes of MRAs, culminating in the emergence of finerenone as a novel therapeutic option for high-risk CKD populations, particularly those with type 2 diabetes mellitus (T2DM).

## 1. Introduction

Heart failure (HF) is a common disease in industrialized countries, and it affects more than 56.2 million people worldwide with a prevalence of 1–2% of adults, which increases with age: from around 1% for those aged <55 years to >10% in those aged 70 years or over [1,2]. Nevertheless, studies usually include patients with overt HF diagnosis, and the real prevalence in the general population is likely to be higher [3]. Currently, the incidence of HF in Europe is about 3/1000 person-years (all age groups), or about 5/1000 person-years in adults [2,4]. HF remains a leading global cause of mortality, morbidity, and poor quality of life (QoL) due to its high use of resources and healthcare costs, making it an area of active research [3].

Throughout the years different definitions and classifications for heart failure (HF) were proposed: the clinical consensus of 2021, issued by the European society of Cardiology and associations of cardiology worldwide, defined HF as a clinical syndrome with symptoms and signs caused by a structural and functional cardiac abnormality and corroborated by elevated natriuretic peptide levels and objective evidence of pulmonary or systemic congestion [5]. One of the most applied classifications is based on the left ventricular ejection fraction (LVEF): an EF of >50% configures a preserved EF HF (HFpEF), while a decreased EF of <40% defines a reduced EF HF (HFrEF); an EF between 40 and 49% represents an HF with mild reduced ejection fraction (HFmEF). However, in recent years new definitions were introduced: HF with improved EF concerns an EF improvement over time of >10% in patients with a baseline EF measurement of <40%; more recently a new phenotype related to patients who are frequent flyers has been identified with the term worsening HF: the current definition refers to episodes of clinical deterioration, often requiring repeated hospitalizations, in the absence of a significant decline in cardiac function or functional capacity [6,7].

CKD is a frequent condition affecting HF, and the two condition are often associated: CKD affects approximately 40–50% of HF patients, particularly in more advanced stages; however, renal dysfunction may be present even in more stable conditions; this is due to shared risk factors like diabetes, hypertension, and metabolic disorders. CKD can worsen cardiovascular function by contributing to hypertension, vascular calcification due to increased workload, and vascular stiffness; otherwise, HF can impair renal function via neurohormonal and inflammatory activation, increased venous pressure, and hypoperfusion [8]. Oxidative stress and fibrosis likely play central roles in the pathophysiology of HF with CKD. Although CKD and worsening renal function are more prevalent in HFpEF compared to HFmrEF and HFrEF, they appear to have less impact on outcomes in HFpEF [9]. CKD remains a major independent predictor of increased mortality and morbidity in HF [10]; however, not all changes in renal markers indicate poor outcomes. For example, when initiating RAAS inhibitors, angiotensin receptor–neprilysin inhibitors (ARNIs), or sodium–glucose cotransporter-2 (SGLT2) inhibitors, a transient drop in the glomerular filtration rate (GFR) and a rise in serum creatinine may occur due to reduced intraglomerular pressure. These changes are typically reversible and occur alongside long-term renal and cardiovascular benefits [11]. Increases in serum creatinine of <50% (if <266 μmol/L or 3 mg/dL) or eGFR declines <10% (if eGFR remains >25 mL/min/1.73 m^2^) are generally acceptable; similarly, transient increases in creatinine during acute HF treatment with diuretics are not linked to worse outcomes if congestion is resolved [12].

## 2. The Current Role of MRA in HF Treatment

Mineralocorticoid receptor antagonists (MRAs) are endorsed among the four recognized treatment pillars, and much evidence revealed beneficial effects in patients with HFrEF and, more recently, in selected individuals with HFpEF. The therapeutic rationale behind MRA use is grounded in the pathophysiology of the renin–angiotensin–aldosterone system (RAAS), which plays a central role in the maintenance of cardiovascular and renal homeostasis [13,14].

In the context of HF, the chronic activation of RAAS—characterized by elevated levels of angiotensin II and aldosterone—contributes to maladaptive cardiac remodeling and disease progression.

Aldosterone, a key effector of RAAS, is primarily secreted by the adrenal cortex in response to angiotensin II, hyperkalemia, and corticotropin. However, extra-adrenal production by endothelial cells, vascular smooth muscle cells, and even cardiomyocytes has also been documented, suggesting local tissue-specific effects [15,16]. Excess aldosterone promotes myocardial hypertrophy, fibrosis, vascular inflammation, endothelial dysfunction, and apoptosis, thereby accelerating adverse cardiac remodeling and increasing susceptibility to arrhythmias and cardiovascular events [17]. Additionally, the phenomenon of aldosterone escape—whereby aldosterone levels rise despite upstream RAAS blockade with ACE inhibitors or ARBs—provides a strong pathophysiological rationale for direct mineralocorticoid receptor (MR) inhibition [18]. Several large-scale randomized controlled trials have confirmed the clinical efficacy of MRAs in HF management (Figure 1). MRAs are capable of reducing mortality and hospitalization in CHF of an ischemic nature. These findings have been confirmed in patients with HFrEF and mild symptoms, solidifying the role of MRAs as guideline-directed medical therapy (GDMT) [19]. In this framework, the TOPCAT trial evaluated spironolactone in patients with HFpEF, a population historically lacking evidence-based therapy. While the overall results were mixed, a regional subgroup analysis—particularly in the Americas—suggested a potential benefit, though the risk of hyperkalemia and worsening renal function remained concerns, especially in those with lower baseline eGFR [20].

Together with cardiovascular effects, MRAs have shown promising findings in cardiorenal protection, particularly in patients with coexisting CKD and T2DM, where aldosterone-mediated injury is especially pronounced. The advent of non-steroidal MRAs like finerenone marks a new chapter in MRA therapy. In the FIDELIO-DKD and FIGARO-DKD trials, finerenone demonstrated consistent cardiovascular and renal benefits, leading to its approval in patients with T2DM and CKD [21,22]. Although the primary focus of these trials was kidney outcomes, the cardiovascular findings support the expanding role of MRAs in the HF population, particularly where comorbid metabolic or renal disease is present.

## 3. Specific MRA Trial Results

To critically examinate the main findings of MRA treatment, a detailed report on patients’ characteristics, HF etiology, and study endpoints may be of relevance. Accordingly, every cited study evaluated specific aspects, providing distinct contributions for elucidating the therapeutic role of MRAs in HF management.

The Randomized Aldactone Evaluation Study (RALES) was a landmark trial that provided important insights into the management of severe HFrEF. The study enrolled 1663 patients with NYHA class III or IV HF and an LVEF of ≤35%, all of whom were receiving standard therapy including ACE inhibitors and loop diuretics. The trial’s primary endpoint was all-cause mortality, and it was prematurely stopped due to the overwhelming benefit observed in the active arm. Patients treated with spironolactone experienced a 30% relative reduction in mortality (relative risk, 0.70; 95% CI, 0.60–0.82; *p* < 0.001) and a 35% reduction in hospitalizations for worsening HF (RR, 0.65; 95% CI, 0.54–0.77; *p* < 0.001) [23]. Beyond mortality benefits, spironolactone therapy was linked to reductions in myocardial fibrosis markers such as procollagen type I and type III amino terminal peptides (PINP and PIIINP), suggesting decreased pathological remodeling of the cardiac extracellular matrix, which correlated with improved patient outcomes [24]. Moreover, neurohormonal improvements were noted with a 23% reduction in brain natriuretic peptide (BNP) plasma concentrations at six months, reflecting enhanced left ventricular diastolic function and reduced filling pressures [25,26].

Despite these profound clinical benefits, the RALES had limitations that affected its real-world application. Notably, less than 10% of patients were on beta blockers at baseline, a standard therapy now known to improve HF outcomes. Furthermore, spironolactone’s hormonal side effects—gynecomastia or mastodynia—were reported in approximately 10% of men, and the risk of hyperkalemia, although low within the trial, became a major concern in clinical practice, especially when combined with ACE inhibitors. These safety issues limited its broader uptake and highlighted the need for MRAs with improved tolerability and safety profiles [23,27]. The success of the RALES thus established MRAs as a cornerstone in HFrEF treatment while driving further research into agents like finerenone, aiming to mitigate side effects without compromising efficacy (Table 1).

The TOPCAT trial was a large, randomized, placebo-controlled study designed to evaluate the efficacy of spironolactone in patients with HFpEF. It enrolled 3445 symptomatic patients with an LVEF of ≥ 45%, who either had a documented hospitalization for HF within the previous year or elevated natriuretic peptide levels (BNP > 100 pg/mL or NT-proBNP > 360 pg/mL). The primary composite endpoint encompassed cardiovascular death, hospitalization for HF, or sudden death. Although the trial did not demonstrate a statistically significant reduction in this primary outcome in the overall study population, a significant decrease in HF hospitalizations was observed in the spironolactone group compared to the placebo group (12.0% vs. 14.2%; hazard ratio [HR]: 0.83; 95% confidence interval [CI]: 0.69–0.99; *p* = 0.042). The subsequent analysis revealed significant geographical discrepancies and race heterogeneity, with patients from the Americas exhibiting more favorable clinical responses compared to those from Eastern Europe. This regional disparity raised concerns that patients recruited primarily based on prior hospitalization in Eastern Europe may have had a lower intrinsic risk or were potentially misclassified, thereby attenuating the overall treatment effect. Spironolactone treatment was associated with a significantly higher incidence of hyperkalemia (serum potassium ≥ 5.5 mmol/L: 18.7% vs. 9.1%; *p* < 0.001) and a greater frequency of serum creatinine doubling relative to the placebo. These findings underscore the complexity of targeting MRAs in HFpEF and highlight the importance of patient selection and regional variation in clinical trial outcomes [20] (Table 1).

The EPHESUS and EMPHASIS-HF trials significantly expanded the clinical role of eplerenone, a selective MRA, in HF management, demonstrating efficacy in distinct patient populations with left ventricular dysfunction.

The Eplerenone Post-Acute Myocardial Infarction (MI) Heart Failure Efficacy and Survival Study (EPHESUS) was conducted in the post-RALES era to evaluate eplerenone in 6632 patients enrolled 3 to 14 days after acute MI who exhibited LVEF ≤ 40% alongside clinical HF or diabetes mellitus. The trial’s two co-primary endpoints were all-cause mortality and a composite of cardiovascular death and hospitalization for cardiovascular causes. Over a mean follow-up of 16 months, eplerenone conferred a 15% relative risk reduction in all-cause mortality (14.4% vs. 16.7%; RR 0.85; 95% CI, 0.75–0.96; *p* = 0.008) and a 13% relative risk reduction in the composite endpoint (26.7% vs. 29.6%; RR 0.87; 95% CI, 0.79–0.95; *p* = 0.002). Cardiovascular death was also significantly lowered (12.3% vs. 14.6%; RR 0.83; 95% CI, 0.72–0.94; *p* = 0.005), with an early and notable 21% reduction in sudden cardiac death (RR 0.79; *p* = 0.03). Importantly, approximately 85% of participants were concurrently treated with beta blockers, addressing a major limitation of the RALES and validating the efficacy of MRAs within contemporary, comprehensive HF regimens [28]. Safety data reflected eplerenone’s greater receptor selectivity, with fewer hormonal adverse effects such as gynecomastia compared to spironolactone, although a modest, persistent decline in estimated glomerular filtration rate (eGFR) was observed early on, and this decline sustained throughout the treatment [29].

Subsequently, the EMPHASIS-HF trial investigated eplerenone’s efficacy in patients with milder chronic HF symptoms (NYHA class II) and an LVEF of ≤30% (or up to 35% with prolonged QRS duration). Enrolling 2737 patients aged 55 or older, EMPHASIS-HF assessed a primary composite outcome of cardiovascular death or first hospitalization for HF. The trial was terminated early after a median 21-month follow-up due to clear evidence of its benefit, with eplerenone reducing the primary endpoint by 37%. Significant improvements were also observed in all-cause mortality (12.5% vs. 15.5%; HR: 0.76; *p* = 0.008), cardiovascular death (10.8% vs. 13.5%; HR: 0.76; *p* = 0.01), and HF hospitalizations (12.0% vs. 18.4%; HR: 0.58; *p* < 0.001). Additionally, a sub-analysis demonstrated a significant reduction in new-onset atrial fibrillation or flutter with eplerenone treatment (2.7% vs. 4.5%; HR: 0.58; *p* = 0.034). The safety profile showed a higher incidence of hyperkalemia (11.8% vs. 7.2%; *p* < 0.001) but similar rates of renal impairment compared to the placebo. However, the trial’s early termination likely led to an underestimation of adverse event incidence [24,30].

Collectively, EPHESUS and EMPHASIS-HF solidified the role of eplerenone across a broader spectrum of left ventricular systolic dysfunction—from early post-MI patients to those with mild chronic HF symptoms—supporting its inclusion as a cornerstone therapy in modern HF guidelines. Despite its improved tolerability relative to spironolactone, hyperkalemia remains the principal safety concern, underscoring the ongoing need for novel MRAs with enhanced safety profiles, such as finerenone [5] (Table 1).

Finerenone is a recently introduced, selective, non-steroidal mineralocorticoid receptor antagonist (MRA) developed to address some of the limitations associated with traditional steroidal MRAs. Unlike its predecessors, such as spironolactone and eplerenone, finerenone has a distinct non-steroidal chemical structure, which enables a unique mode of interaction with the mineralocorticoid receptor. It demonstrates high affinity and selectivity for the mineralocorticoid receptor while exhibiting minimal activity at other hormone receptors, including androgen, progesterone, glucocorticoid, and estrogenic receptors. This receptor specificity reduces the risk of hormonal side effects commonly seen with steroidal MRAs, such as gynecomastia and menstrual irregularities [31,32] (Table 1).

Pharmacokinetically, finerenone has a relatively short half-life and does not generate active metabolites. It exhibits a more balanced distribution between cardiac and renal tissues compared to steroidal MRAs, which may contribute to improved cardioprotective effects and reduced renal adverse events [33]. This tissue distribution profile, combined with its pharmacodynamics, suggests a more favorable risk–benefit ratio and a potentially lower incidence of clinically significant hyperkalemia [34].

These theoretical advantages were substantiated in two early-phase studies such as ARTS and ARTS-HF, which supported the tolerability and safety of finerenone in patients with HF and CKD [35,36]. From this preliminary analysis and positive background, two large phase 3 interventional trials, FIDELIO-DKD and FIGARO-DKD, evaluated the efficacy and safety of finerenone in patients with type 2 diabetes (T2D) and CKD.

The FIDELIO-DKD trial enrolled approximately 5700 patients with T2D and advanced CKD. Finerenone significantly reduced the risk of the primary renal composite endpoint—comprising kidney failure, a sustained ≥40% decline in estimated glomerular filtration rate (eGFR), or renal death—by 18% compared to the placebo (HR: 0.82; 95% CI: [0.73–0.93]; *p* = 0.001). A key secondary cardiovascular composite endpoint—CV death, non-fatal myocardial infarction (MI), non-fatal stroke, or hospitalization for HF (HHF)—was also significantly reduced by 14% (HR: 0.86; 95% CI: [0.75–0.99]; *p* = 0.034). Although hyperkalemia incidence was higher in the finerenone group, it was generally manageable with monitoring and dose adjustments. An initial decrease in eGFR was observed, but it plateaued over time [22,37].

Complementing this, the FIGARO-DKD trial included a broader population of ~7400 patients with earlier-stage CKD. The primary endpoint—CV death, non-fatal MI, non-fatal stroke, or HHF—was reduced by 13% (HR: 0.87; 95% CI: [0.76–0.98]; *p* = 0.026), driven largely by a 29% reduction in HHF (HR: 0.71; 95% CI: [0.56–0.90]). However, the key renal composite outcome (a sustained ≥57% decline in eGFR, kidney failure, or renal death) was not significantly improved (HR: 0.93; 95% CI: [0.76–1.15]) [37].

To better understand finerenone’s effects across CKD severity in T2D, the FIDELITY pooled analysis synthesized individual patient-level data from both trials (N > 13,000). Finerenone reduced the composite CV outcome by 14% (HR: 0.86; 95% CI: [0.78–0.95]; *p* = 0.0018) and the composite renal outcome by 23% (HR: 0.77; 95% CI: [0.67–0.88]; *p* = 0.0002). Notably, HHF was reduced by 22% (HR: 0.78; 95% CI: [0.66–0.92]), with additional reductions in sudden cardiac death and both all-cause and CV mortality during on-treatment periods. Hyperkalemia remained the principal safety concern but was generally well controlled [21].

In the HF domain, the FINEARTS-HF trial marked a significant advancement by evaluating finerenone in patients with symptomatic HFpEF and HFmrEF, where LVEF ≥ 40%. This double-blind, placebo-controlled trial enrolled ~6000 patients and assessed the total number of HF worsening events (hospitalizations or urgent visits requiring IV therapy) and CV death. Finerenone significantly reduced the primary endpoint, primarily by lowering recurrent hospitalizations for worsening HF (RR: 0.82; 95% CI: [0.71–0.94]; *p* = 0.006). However, it did not significantly reduce CV mortality as a standalone outcome [38].

Additional findings from FINEARTS-HF underscored finerenone’s renal effects in a population at a relatively low renal risk. While the composite kidney outcome (a ≥50% eGFR decline or kidney failure) was numerically higher (HR: 1.33; 95% CI: [0.94–1.89]), finerenone significantly reduced the development of microalbuminuria and macroalbuminuria, consistent with its known antiproteinuric effects [38]. Importantly, the drug improved patient-reported outcomes (Kansas City Cardiomyopathy Questionnaire scores) and showed a consistent benefit across different LVEF subgroups, highlighting its relevance in HFpEF—a syndrome with few effective therapeutic options. Hyperkalemia incidence was elevated but lower than typically observed with spironolactone in prior trials such as TOPCAT (Table 1 and Figure 2).

The integration of finerenone into clinical practice has been reflected in evolving guideline recommendations. The 2023 ESC HF guidelines recognized its benefit in patients with HFpEF, especially following the publication of FINEARTS-HF and prior evidence in CKD and T2D populations. Finerenone is now viewed as a valuable addition to the therapeutic arsenal across the HF spectrum, including HFrEF, HFmrEF, and HFpEF [5,39].

Emerging data also suggest a synergistic role for finerenone in combination therapy. Its use alongside SGLT2 inhibitors and traditional renin–angiotensin–aldosterone system (RAAS) blockers may yield incremental benefits by addressing fibrosis, oxidative stress, and glomerular hemodynamic [32,40]. These mechanisms underpin its dual cardiorenal protection, positioning finerenone as a unique agent at the interface of HF and CKD management.

## 4. Evidence of MRA in High-Risk CKD Patients

The role of MRAs in slowing CKD progression has evolved substantially over the past two decades. Initially designed for cardiovascular indications, MRAs have shown potential to mitigate kidney damage through anti-inflammatory and anti-fibrotic mechanisms [41,42].

Early investigations of spironolactone and eplerenone established their efficacy in cardiovascular settings but revealed mixed renal outcomes, particularly in patients with impaired baseline kidney function. In the TOPCAT (Treatment of Preserved Cardiac Function Heart Failure with an Aldosterone Antagonist) trial, which enrolled patients with HFpEF, a large subset of whom had CKD, spironolactone did not reduce the primary composite cardiovascular endpoint in the overall population. In regional subgroup analyses, particularly in patients from the Americas, cardiovascular benefits were more pronounced [43]. However, spironolactone was associated with an increased incidence of hyperkalemia and worsening renal function, especially among those with lower eGFR at baseline, highlighting the challenge of safely using steroidal MRAs in CKD populations [29,43,44].

Similarly, in the EPHESUS trial, which assessed eplerenone in patients with HF post-myocardial infarction, a transient decline in eGFR was observed in the eplerenone arm. Despite this apparently unfavorable effect on renal function, the favorable cardiovascular outcome in the active arm was reached. Current findings highlight the importance of monitoring repetitive laboratory measurements in subjects with potential nephrotoxicity risk and patients with more advanced renal dysfunction and reduced renal reserve, raising some caution about long-term use in severe CKD [24,27].

The development of finerenone, a non-steroidal and more selective MRA, addressed many limitations related to steroidal MRAs. Preclinical data demonstrated its ability to inhibit pro-inflammatory and pro-fibrotic signaling with minimal effect on serum potassium homeostasis and less activation of epithelial sodium channels [31,32,45].

The landmark FIDELIO-DKD trial enrolled 5734 patients with T2DM and CKD (an eGFR of 25 to <75 mL/min/1.73 m^2^, all with albuminuria). Finerenone significantly reduced the primary kidney composite endpoint—defined as kidney failure, a sustained ≥40% decline in eGFR, or renal death—by 18% (HR, 0.82; 95% CI, 0.73–0.93; *p* = 0.001). Additionally, it showed a statistically significant reduction in the incidence of end-stage kidney disease (ESKD) [22]. Importantly, finerenone also led to a 31% reduction in UACR at 4 months, underscoring its anti-fibrotic renal activity [21] (Table 2).

The FIGARO-DKD trial, which enrolled 7437 patients with a broader spectrum of CKD (an eGFR of 25 to 90 mL/min/1.73 m^2^), had a primary cardiovascular focus, but renal outcomes were assessed as key secondary endpoints. The primary kidney composite outcome (a ≥40% eGFR decline, kidney failure, or renal death) showed a non-significant trend toward benefits. However, a more stringent secondary renal composite endpoint—defined as a ≥57% sustained eGFR decline, kidney failure, or renal death—demonstrated a statistically significant improvement with finerenone, including a reduced incidence of ESKD [46] (Table 2).

These findings were consolidated in the FIDELITY pooled analysis, combining patient-level data from both FIDELIO-DKD and FIGARO-DKD (n = 13,026). The analysis confirmed a 23% reduction in the composite kidney outcome (a ≥57% eGFR decline, kidney failure, or renal death; HR, 0.77; 95% CI, 0.67–0.88) and a 20% reduction in the risk of kidney failure alone [21]. These renoprotective effects were consistent across a range of baseline eGFR and UACR levels, reinforcing finerenone’s utility in both early and advanced diabetic CKD [21,46,47] (Table 2).

Post hoc subgroup analyses and real-world extensions have further validated the renal benefits of finerenone in diverse and traditionally underrepresented groups. For example, the FINARTS study demonstrated that finerenone effectively reduced albuminuria and blood pressure in patients with resistant hypertension and CKD, without increasing the incidence of hyperkalemia beyond manageable thresholds [48]. Similarly, Lerma et al. confirmed that finerenone conferred renal protection in elderly CKD patients, reducing the risk of renal dysfunction progression and electrolyte abnormalities [49]. Moreover, data from Mentz et al. indicated that even in patients with preserved eGFR but high UACR, finerenone slowed renal function decline, suggesting a rationale for earlier intervention before irreversible nephron loss [50].

From the mixed renal outcomes seen with steroidal MRAs to the consistent kidney-protective effects observed with finerenone, the therapeutic landscape for MRAs in CKD has dramatically evolved. Finerenone has emerged as a pivotal treatment in diabetic kidney disease, offering robust evidence for slowing disease progression and reducing ESKD risk across a broad spectrum of renal function. This marks a paradigm shift in the application of MRA therapy—positioning it not only as a cardioprotective agent but also as a cornerstone of renoprotection in high-risk CKD populations [21,51].

## 5. Biochemical, Pharmacological, and Cellular Effects of Finerenone in Cardiorenal Disease

Mineralocorticoid receptor antagonists (MRAs) have gained increasing attention for their role in targeting the downstream effects of renin–angiotensin–aldosterone system (RAAS) overactivation in cardiovascular and renal disease. Among them, finerenone has emerged as a distinct, non-steroidal MRA with unique molecular, pharmacological, and clinical properties that differentiate it from traditional steroidal agents.

Upon binding to the mineralocorticoid receptor (MR), finerenone induces a specific conformational change that prevents the recruitment of critical transcriptional co-regulators, such as steroid receptor coactivator-1 (SRC-1) and other nuclear cofactors, which are essential for the full transcriptional activation of pro-inflammatory and pro-fibrotic genes [32,52]. This mechanism sets finerenone apart from steroidal MRAs like spironolactone and eplerenone, which may act as partial agonists at the MR and still allow partial cofactor recruitment [53]. Finerenone functions as an inverse agonist, meaning that it can suppress baseline MR activity even in the absence of aldosterone, thereby further limiting the expression of pathogenic gene pathways [54,55]. Aldosterone, the main endogenous ligand of the MR, typically facilitates MR nuclear translocation and transcriptional activation of genes related to sodium retention, inflammation, and fibrosis. Finerenone effectively inhibits this nuclear import process, directly blocking the aldosterone-mediated transcriptional response [51]. This results in a gene expression profile that is not only less pro-inflammatory and pro-fibrotic than that induced by steroidal MRAs but also more protective under conditions of elevated aldosterone, cortisol, or corticosterone—hormonal states often present in advanced disease [56,57].

These molecular distinctions translate into important downstream effects at the cellular and tissue levels. A central feature of MR overactivation is the amplification of oxidative stress, particularly through the upregulation of nicotinamide adenine dinucleotide phosphate (NADPH) oxidases such as NOX2 and NOX4. These enzymes contribute to excessive production of reactive oxygen species (ROS), which in turn cause DNA damage and propagate redox-sensitive signaling cascades [58,59]. Finerenone has been shown to inhibit NOX activity, reduce ROS generation, and consequently attenuate oxidative stress-driven activation of nuclear factor kappa-light-chain-enhancer of activated B (NF-κB) cells, a pivotal transcription factor in inflammatory gene expression [34,45].

In parallel, finerenone exerts powerful anti-inflammatory effects by downregulating the expression of pro-inflammatory cytokines and chemokines, including TNF-α, IL-1β, and CCL-2. This is achieved through transcriptional repression mediated by MR antagonism, which disrupts the molecular signaling pathways that typically exacerbate immune cell activation and tissue inflammation [56,60]. Preliminary data suggest that finerenone may also modulate the phenotype and function of immune cells by altering the tissue sodium environment, which has been linked to pro-inflammatory cellular states [34,45,61].

The anti-fibrotic actions of finerenone are equally significant. Fibrosis is a hallmark of both progressive chronic kidney disease (CKD) and heart failure (HF), characterized by excess deposition of extracellular matrix components. Finerenone downregulates pro-fibrotic genes such as transforming growth factor-beta 1 (TGF-β1), connective tissue growth factor (CTGF), plasminogen activator inhibitor-1 (PAI-1), fibronectin, and various collagen isoforms [56,62,63,64]. In preclinical and clinical models, finerenone reduces the pathological accumulation of myofibroblasts—the primary matrix-producing cells—and collagen deposition in both renal and cardiac tissues. These effects occur regardless of changes in systemic blood pressure, indicating a direct anti-fibrotic mechanism [65]. In the myocardium, finerenone limits interstitial fibrosis and improves ventricular compliance, which may help counter diastolic dysfunction and HFpEF progression [66]. In the kidney, the drug preserves the glomerular architecture and mitigates tubulointerstitial fibrosis, thereby slowing the progression of CKD [56,62] (Figure 3).

Finerenone’s clinical utility is further enhanced by its favorable pharmacokinetic characteristics. It has a short plasma half-life of approximately 2–3 h and does not produce active metabolites, resulting in predictable drug exposure and reduced potential for drug accumulation [34]. This is especially relevant in patients with reduced renal clearance, a population commonly treated with MRAs. Furthermore, finerenone exhibits even tissue distribution between renal and cardiac compartments, avoiding the renal-biased accumulation seen with steroidal MRAs [45]. This balanced biodistribution may contribute to its ability to offer simultaneous protection across organ systems [21,46]. Importantly, with less than 1% of the parent compound excreted renally, the drug’s pharmacokinetics are relatively stable even in patients with moderate to advanced renal impairment. This minimizes the risk of hyperkalemia, a frequent and potentially limiting adverse effect observed with traditional MRAs [34,56,67].

Collectively, these pharmacodynamic and pharmacokinetic advantages translate into meaningful clinical benefits. Finerenone’s ability to selectively and potently inhibit MR overactivation—while maintaining a lower risk of endocrine side effects and hyperkalemia—positions it as a powerful therapeutic option for patients with coexisting heart failure and CKD. Its efficacy has been demonstrated in large-scale randomized aldostcontrolled trials, such as FIDELIO-DKD and FIGARO-DKD, which showed significant reductions in both renal and cardiovascular events [21]. Importantly, finerenone may also complement other guideline-directed therapies, including SGLT2 inhibitors and ARNIs, paving the way for integrated approaches to cardiorenal protection [62,68,69,70]. As evidence continues to grow, finerenone is expected to play an increasingly prominent role in the management of cardiorenal syndromes, offering a targeted and well-tolerated strategy to reduce disease progression and improve outcomes [21,56].

## 6. Conclusions and Future Perspective

The emerging role of finerenone compared to traditional MRA treatment is well demonstrated particularly in patients with CKD and HFpEF. Its well-known anti-fibrotic and anti-oxidative stress properties drive cardiac remodeling and endothelial vascular damage reduction. Despite studies initially demonstrated this evidence in patients with HFrEF, recent data suggest that beneficial advantages could be extended in HFrEF. Additional findings suggest that the simultaneous occurrence of two conditions may benefit from this treatment. Although positive effects are probably due to class specificity with common effects in all MRA drugs, some evidence suggests that additive beneficial effects are probably related to finerenone receptor affinity which may provide specific benefit to kidney and cardiac metabolism and function, beyond the effects of RAASi and beta blockers. Moreover, the higher selectivity for MR revealed by finerenone potentially lowers the risk of hyperkalemia and anti-androgenic side effects. Current findings support the inclusion of finerenone in the treatment regimen for appropriate patients within the HFpEF and HFmEF subgroups.

Despite the drug’s efficacy being known in CKD and HFpEF, little evidence exists for patients affected by HFrEF and advanced renal dysfunction with GFR < 30 mL/min/m^2^. In these settings finerenone administration requires further investigations; indeed, in FIDELIO and Figaro the percentage of patients with reduced ejection fraction was negligible, and FINEARTS investigated patients with LVEF > 40%. Additionally, finerenone in the retrospective analysis did not demonstrate an additive benefit compared to eplerenone or spironolactone; the introduction of the drug on top of traditional treatment may not translate to improved outcomes, and it may increase the risk of hyperkalemia without improved outcomes.

Similarly, safety and efficacy in patients with advanced CKD are not established since Fidelio excluded patients with GFR < 25 mL/min/m^2^ and subjects undergoing dialytic treatment. According to the current literature, finerenone administration is recommended in pt with mildly reduced preserved EF and those with stages 1–3 CKD. Further studies may investigate the role of the drug in patients with reduced LVEF by a prespecified head-to-head study to establish the risk of hyperkalemia and eventual the additive role of the drug compared to spironolactone and eplerenone. Similarly in the severe CKD stage of non-diabetic etiology such as hypertensive nephropathy, glomerulonephritis, or polycystic kidney disease, the role of finerenone remains to be established. Future studies will investigate the role of finerenone in patients with HFrEF, in those with more severe CKD stages, and in subjects with advanced HF associated with low cardiac output and blood pressure profiles.

## Figures and Tables

**Figure 1 biomedicines-13-01693-f001:**
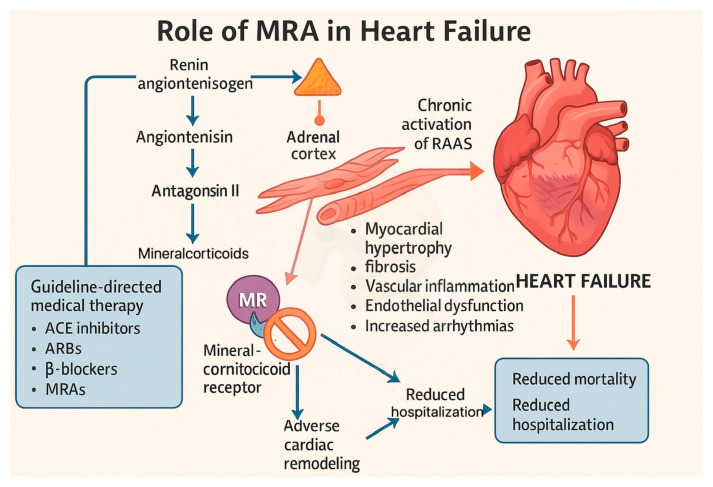
This figure illustrates the role of mineralocorticoid receptor antagonists (MRAs) in heart failure. The activation of the renin–angiotensin–aldosterone system (RAAS) leads to aldosterone release, which binds to mineralocorticoid receptors (MRs), promoting myocardial hypertrophy, fibrosis, vascular inflammation, and adverse cardiac remodeling—ultimately contributing to heart failure. MRAs block this pathway, reducing cardiac remodeling, hospitalizations, and mortality when used as part of guideline-directed medical therapy (GDMT). MRA, Mineralocorticoid Receptor Antagonist; RAAS, Renin–Angiotensin–Aldosterone System; MR, Mineralocorticoid Receptor; ACE, Angiotensin-Converting Enzyme; ARB, Angiotensin Receptor Blocker; GDMT, Guideline-Directed Medical Therapy.

**Figure 2 biomedicines-13-01693-f002:**
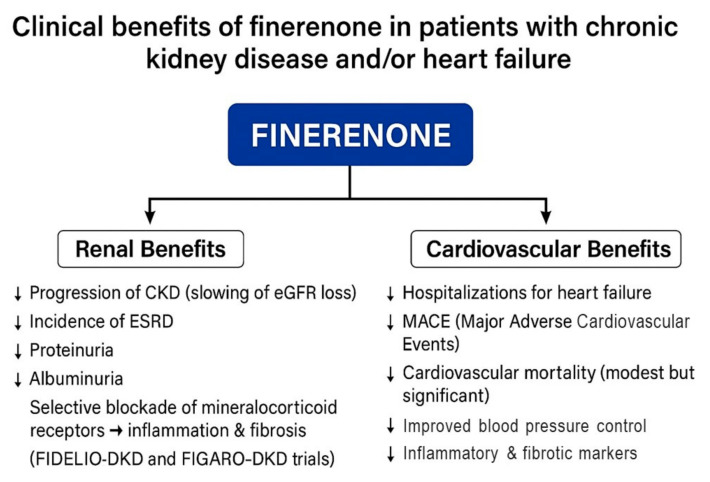
Clinical benefits of finerenone in patients with chronic kidney disease and/or heart failure. CKD, chronic kidney disease; eGFR, estimated glomerular filtration rate; ESRD, end-stage renal disease; MACE, major adverse cardiovascular events (a composite of cardiovascular death, non-fatal myocardial infarction, and non-fatal stroke); FIDELIO-DKD, Finerenone in Reducing Kidney Failure and Disease Progression in Diabetic Kidney Disease; FIGARO-DKD, Finerenone in Reducing Cardiovascular Mortality and Morbidity in Diabetic Kidney Disease.

**Figure 3 biomedicines-13-01693-f003:**
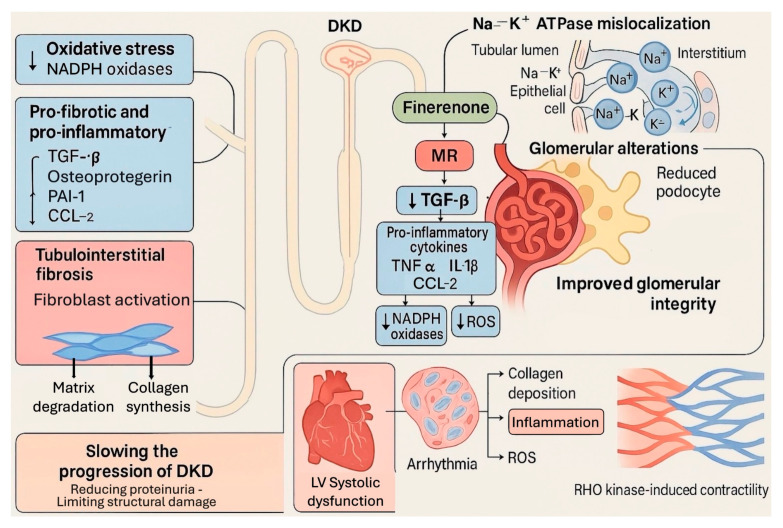
Mechanisms of finerenone in diabetic kidney disease (DKD). Finerenone blocks mineralocorticoid receptor (MR) overactivation, reducing pro-inflammatory and pro-fibrotic signaling, oxidative stress, and Na^+^/K^+^-ATPase mislocalization. These effects lead to improved glomerular integrity, reduced tubulointerstitial fibrosis, and slower DKD progression. In the heart, finerenone mitigates inflammation, fibrosis, and oxidative stress, contributing to improved cardiac function and reduced arrhythmic risk. CCL-2, C-C motif chemokine ligand 2; DKD, diabetic kidney disease; IL-1β, interleukin-1 beta; LV, left ventricular; MR, mineralocorticoid receptor; NADPH, nicotinamide adenine dinucleotide phosphate; PAI-1, plasminogen activator inhibitor-1; ROS, reactive oxygen species; TGF-β, transforming growth factor beta; TNF-α, tumor necrosis factor alpha.

**Table 1 biomedicines-13-01693-t001:** Baseline characteristics of patients enrolled in major randomized controlled trials in heart failure with reduced ejection fraction (HFrEF). Data are presented from the RALES, EMPHASIS-HF, EPHESUS, TOPCAT, ATHENA, and FINEARTS studies, including demographics, clinical parameters, comorbidities, NYHA class distribution, renal function (eGFR), left ventricular ejection fraction (LVEF), and natriuretic peptide levels (NT-proBNP). Values are expressed as the mean (standard deviation) or number (percentage) where appropriate. “NR” indicates data not reported. The studies vary in sample size, sex distribution, comorbidity burden, and severity of heart failure, reflecting differences in inclusion criteria and patient populations.

Study Name Authors	Rales Vardeny et al., 2021	Emphasis-hf Rossignol et al., 2013	Ephesus Pitt et al., 2003	Topcat Beldhuis et al., 2021	Athena Greene et al., 2019	Finearts Solomon et al., 2024
N. of female male patients	1663 446 (27%) 1217 (73%)	2737 610 (22%) 2127 (78%)	6642 1918 (29%) 4714 (71%)	3445 1775 (52%) 1670 (48%)	360 129 (35%) 241 (65%)	6001 2732 (46%) 3269 (54%)
NYHA CLASS						
I or II III or IV Missing	NR	2730 (100%) 3 (<1%) 4	NR	2303 (67%) 1136 (33%) 6	60 (17%) 302 (83%)	4146 (69%) 1854 (31%) 1
EGFR (mL/min × 1.73 m^2^)	63 (22)	65 (18)	78 (14)	65 (19)	56.5 (16)	63 (20)
Ejection fraction (FE)%	25 (7)	26 (5)	33 (6)	57 (7)	33 (13)	53 (8)
Previous HF hospitalization						
Yes	NR	1438 (53%)	497 (7%)	2489 (72%)	334 (93%)	3619 (60%)
Missing	1663	3	0	3	0	0
NT-proBNP	NR	NR	NR	8430 (4630–17,200)	4102 (2204–8752)	10,414 (4485–19,459)
Diabetes	369 (22%)	859 (31%)	2125 (32%)	1118 (32%)	146 (41%)	2454 (41%)
Hypertension	391 (24%)	1819 (66%)	4051 (61%)	3147 (91%)	301 (84%)	5325 (89%)
Atrial fibrillation	183 (11%)	844 (31%)	NR	1214 (35%)	172 (48%)	3273 (55%)
Myocardial infarction	472 (28%)	1380 (50%)	1793 (27%)	893 (26%)	103 (29%)	1541 (26%)

**Table 2 biomedicines-13-01693-t002:** Baseline demographic and clinical characteristics of patients enrolled in the FIDELIO-DKD, FIGARO-DKD, and pooled FIDELITY trials. The table summarizes age, sex, renal function (eGFR and UACR), cardiovascular risk factors, blood pressure, glycemic control (HbA1c), serum potassium, diabetes duration, and background therapy at baseline. Data are reported as the mean (±standard deviation) or percentage, except for UACR values, which are presented as medians. The population is characterized by a high prevalence of established cardiovascular disease and the widespread use of renin–angiotensin system blockers and statins.

Patients Characteristic	FIDELIO-DKD	FIGARO-DKD	FIDELITY
AGE (years)	66.6 (±9.1)	64.1 (±9.8)	64.8 (±9.5)
MALE (%)	69.8%	69.4%	69.8%
eGFR (mL/min/1.73m^2^)	44.3 (±12.6)	67.8 (±21.7)	57.6 (±17.1)
UACR (mg/g)	852 (median)	312 (median)	515 (median)
Systolic blood pressure (mmHg)	136.7 (±14.2)	136 (±14)	136.7 (±14.2)
HbA1c (%)	7.7 (±1.4)	7.7 (±1.4)	7.7 (±1.4)
Serum potassium (mmol/L)	4.3 (±0.4)	4.3 (±0.4)	4.3 (±0.4)
Diabetes duration (years)	15.4 (±8.6)	14.5 (±8.5)	15.0 (±8.6)
History of CV disease (%)	45.6%	44.3%	45.6%
ACEi/ARB use (%)	99.8%	99.7%	99.8%
Statins use (%)	72.2%	70.3%	72.2%
Diuretic use (%)	51.5%	47.4%	51.5%

## Data Availability

No new data were created or analyzed in this study. Data sharing is not applicable to this article.

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
