# Peer review of "Current Application of Mineralocorticoid Antagonist (MRA) in Heart Failure and CKD: Does Non-Steroidal Drug Add Novel Insights"

_biomedicines, 2025, doi:10.3390/biomedicines13071693_

Round 1

Reviewer 1 Report

Comments and Suggestions for Authors

This is a well written review paper covering the most important aspects of MRA in HF and CKD with focus on finerenone as a novel treatment option. It has the potential to contribute to the overall knowledge regarding this research area.

I have only minor comments in attached file that I suggest to revise before publication. 

For example:

  • change spelling from hyperkaliemia to hyperkalemia
  • quite a few paragraphs lack references
  • tables and figures - please explain abbreviations and revise misspellings
  • in page 8: FIDELIO-DKD and FIGARO-DKD trial results are repeated from page 6 but different references are used. Which references are correct?
  • in the conclusion, I suggest to omitt the sentense: "Combining finerenone with SGLT2 inhibitors appears to be safe and potentially aynergistic..." This is only discussed in a short paragraph in the paper and not central enough to be a such detailed part of the conclusion.

Author Response

Comment: This is a well-written review paper covering the most important aspects of MRA in HF and CKD with focus on finerenone as a novel treatment option. It has the potential to contribute to the overall knowledge regarding this research area. I have only minor comments in the attached file that I suggest to revise before publication.

Response: Thank you for your kind words and encouraging comments. We appreciate your positive assessment of the manuscript's value and potential contribution to the field. We have addressed all the suggested revisions in detail below.

Comment: Change spelling from hyperkaliemia to hyperkalemia.

Response: Thank you for pointing this out. We agree with this comment, therefore, we have corrected the spelling from hyperkaliemia to hyperkalemia in the revised manuscript and other spelling errors kindly pointed out.

Comment: Quite a few paragraphs lack references.

Response: Thank you for this valuable observation. We agree with this comment.
Therefore, we have reviewed the manuscript and added references where relevant statements previously lacked citations.

Notable additions include:

Section 2: We have added several references to better highlight the role and mechanisms of mineralocorticoids in heart failure (references 14, 16), as well as the therapeutic role of MRAs (reference 19).

Section 3: Additional references were included to support preliminary data (references 26, 27) and to address subgroup analyses in specific populations, such as patients with atrial fibrillation or left ventricular dysfunction after myocardial infarction (references 28, 30). Furthermore, we included references exploring the molecular and pharmacological properties of MRAs (references 31, 32, 33), along with data from the ARTS and ARTS-HF studies (references 35, 36).

Section 4: The newly added references strengthen the manuscript by reinforcing key mechanistic, clinical, and safety aspects. Preclinical studies support the anti-inflammatory and antifibrotic actions of MRAs, particularly highlighting finerenone’s selective MR antagonism with a lower risk of hyperkalemia. New citations addressing the limitations of steroidal MRAs—such as increased hyperkalemia and renal dysfunction—enhance the discussion of earlier trials like TOPCAT and EPHESUS. Recent clinical trial data from FIDELIO-DKD, FIGARO-DKD, and the FIDELITY pooled analysis substantiate finerenone’s consistent renoprotective effects across various stages of CKD and levels of albuminuria. Moreover, real-world studies and subgroup analyses confirm its safety and efficacy in diverse patient groups, including those with preserved eGFR or resistant hypertension, supporting its role as a cornerstone therapy in diabetic CKD.

Section 5: As suggested by another reviewer, we revised this paragraph to include a more detailed molecular explanation of the role of MRAs. Accordingly, we introduced appropriate references to support these mechanistic insights.

Comment: Tables and figures – please explain abbreviations and revise misspellings.

Response: Thank you for pointing this out. We agree with this comment.
Therefore, we have revised all tables and figure legends to include clear definitions of abbreviations and corrected misspellings.

Comment: On page 8: FIDELIO-DKD and FIGARO-DKD trial results are repeated from page 6 but different references are used. Which references are correct?

Response: Thank you for identifying this inconsistency. We agree with this comment.
Therefore, we have revised the references for the FIDELIO-DKD and FIGARO-DKD trial results to ensure consistency and accuracy throughout the manuscript.

Comment: In the conclusion, I suggest omitting the sentence: "Combining finerenone with SGLT2 inhibitors appears to be safe and potentially synergistic..." This is only discussed in a short paragraph in the paper and not central enough to be a such detailed part of the conclusion.

Response:Thank you for this helpful suggestion. We agree with this comment.
Therefore, we have removed the sentence from the conclusion to maintain focus and clarity.

We have greatly appreciated the reviewer’s insightful comments, which have helped improve the quality and clarity of our manuscript. We hope that the revised version meets the expectations of the editorial team and reviewer.

Sincerely,
Irene Carlino

On behalf of all co-authors

Reviewer 2 Report

Comments and Suggestions for Authors
  1. Keywords: Some phrases are long, such as Mineralocorticoid Receptor Antagonists, Type 2 Diabetes Mellitus. As a keyword, it should be refined. Please modify and replace it with synonyms.
  2. Introduction: Too many segments, some paragraphs only have one sentence. It is recommended to integrate paragraphs and keep 3-4 paragraphs.
  3. Section of "2. Current role of MRA in the HF treatment", I suggest adding a diagram to illustrate the role of MRA in the treatment of heart failure.
  4. Table 1: Only the table title is retained above the table, and other explanations are placed below the table. The same issue applies to Table 2.
  5. Tables 1 and 2 need to be cited correctly in appropriate positions within the text content. Also Figure 1 and 2.
  6. Section of "5. Biochemical, Pharmacological and Cellular Effects of Finerenone in Cardiorenal Disease", The depth of writing in this section is insufficient, especially in explaining the processes and mechanisms of biochemistry, pharmacology, cells, and molecules, The author needs to strengthen this part of the content and make major revisions.
  7. "6. Future perspective e conclusion", there is a writing error here.
  8. Revise "Future perspective and conclusion" to "Conclusion and future perspective", Firstly, summarize the current research progress, and then express opinions on future research and challenges based on this. This section can be divided into two paragraphs. The first paragraph summarizes existing research, while the second paragraph proposes future research directions and challenges.
  9. Check if abbreviations are fully included.
  10. Please check the format requirements of the journal, some content is missing, such as author contribution statement, experimental ethics statement, funding information, conflict of interest statement, etc.
  11. As a medical review paper, there are few references cited, and it is necessary to strengthen the analysis of existing research progress and increase the number of references.
  12. The English could be improved to more clearly express the research.

Author Response

Thank you for your kind words and encouraging comments. We appreciate your positive assessment of the manuscript's value and potential contribution to the field. We have addressed all the suggested revisions in detail below.

Comment 1: Keywords: Some phrases are long, such as Mineralocorticoid Receptor Antagonists, Type 2 Diabetes Mellitus. As a keyword, it should be refined. Please modify and replace it with synonyms.

Response 1: Thank you for pointing this out. I agree with this comment.
Therefore, I have revised the keywords to more concise and widely accepted terms.
Revised Keywords:

  • Mineralocorticoid receptor antagonists → MRAs

  • Type 2 Diabetes Mellitus → Type 2 diabetes

These changes can be found on page 1, in the "Keywords" section.

Comment 2: Introduction: Too many segments, some paragraphs only have one sentence. It is recommended to integrate paragraphs and keep 3-4 paragraphs.

Response 2: Thank you for your suggestion.
Therefore, I have revised and merged the introduction into three cohesive paragraphs, improving narrative flow and reducing fragmentation. This revision can be found on page 1 and 2, lines 31–73.

Comment 3: Section of "2. Current role of MRA in the HF treatment", I suggest adding a diagram to illustrate the role of MRA in the treatment of heart failure.

Response 3: Thank you for the helpful suggestion.
Therefore, I have added Figure 1, which illustrates the mechanism and clinical application of MRAs in the treatment of heart failure. This figure is introduced in Section 2, page 3, line 93, and the figure itself is placed on the same page.

Comment 4: Table 1: Only the table title is retained above the table, and other explanations are placed below the table. The same issue applies to Table 2.

Response 4: Thank you for highlighting this formatting issue.
Therefore, I have revised both Table 1 and Table 2 to move all explanatory footnotes and legends below each table.

Comment 5: Tables 1 and 2 need to be cited correctly in appropriate positions within the text content. Also Figure 1 and 2.

Response 5: Thank you for the important remark.
Therefore, I have added proper in-text citations for Table 1, Table 2 and for Figure 1 now named Figure 2, and Figure 2 now named figure 3 at relevant locations.

Comment 6: Section of "5. Biochemical, Pharmacological and Cellular Effects of Finerenone in Cardiorenal Disease": The depth of writing in this section is insufficient... The author needs to strengthen this part...

Response 6: Thank you for this valuable feedback. I agree with this comment.
Therefore, I have significantly revised Section 5 to include detailed explanations of finerenone’s effects on MR nuclear signaling, NADPH oxidase inhibition, transcription factor regulation, and immune modulation. New references were added to support this update.
This revised section appears on pages 10–11, lines 381–456.

Comment 7: "6. Future perspective e conclusion", there is a writing error here.

Response 7: Thank you for catching this typographical error.
Therefore, I have corrected the title to “Conclusion and Future Perspective.”

Comment 8: Revise "Future perspective and conclusion" to "Conclusion and future perspective". Firstly, summarize the current research progress, and then express opinions on future research...

Response 8: Thank you for the structured recommendation.
Therefore, I have revised the section as it appears on page 12-23, lines 470-502.

Comment 9: Check if abbreviations are fully included.

Response 9: Thank you.
Therefore, I have reviewed the manuscript and ensured that all abbreviations are defined at first use and included in the glossary.

Comment 10: Please check the format requirements of the journal, some content is missing, such as author contribution statement, experimental ethics statement, funding information, conflict of interest statement, etc.

Response 10: Thank you for pointing this out.
Therefore, I have added the additional statements as required by journal guidelines.

Comment 11: As a medical review paper, there are few references cited... it is necessary to increase the number of references.

Response 11: Thank you for this valuable observation. We agree with this comment. Therefore, we have reviewed the manuscript and added references where relevant statements previously lacked citations.

Notable additions include:

Section 2: We have added several references to better highlight the role and mechanisms of mineralocorticoids in heart failure (references 14, 16), as well as the therapeutic role of MRAs (reference 19).

Section 3: Additional references were included to support preliminary data (references 26, 27) and to address subgroup analyses in specific populations, such as patients with atrial fibrillation or left ventricular dysfunction after myocardial infarction (references 28, 30). Furthermore, we included references exploring the molecular and pharmacological properties of MRAs (references 31, 32, 33), along with data from the ARTS and ARTS-HF studies (references 35, 36).

Section 4: The newly added references strengthen the manuscript by reinforcing key mechanistic, clinical, and safety aspects. Preclinical studies support the anti-inflammatory and antifibrotic actions of MRAs, particularly highlighting finerenone’s selective MR antagonism with a lower risk of hyperkalemia. New citations addressing the limitations of steroidal MRAs—such as increased hyperkalemia and renal dysfunction—enhance the discussion of earlier trials like TOPCAT and EPHESUS. Recent clinical trial data from FIDELIO-DKD, FIGARO-DKD, and the FIDELITY pooled analysis substantiate finerenone’s consistent renoprotective effects across various stages of CKD and levels of albuminuria. Moreover, real-world studies and subgroup analyses confirm its safety and efficacy in diverse patient groups, including those with preserved eGFR or resistant hypertension, supporting its role as a cornerstone therapy in diabetic CKD.

Section 5: As suggested we revised this paragraph to include a more detailed molecular explanation of the role of MRAs. Accordingly, we introduced appropriate references to support these mechanistic insights.

Comment 12: The English could be improved to more clearly express the research.

Response 12: Thank you.
Therefore, I have revised the manuscript for improved clarity, grammar, and readability.
Revisions span the entire manuscript.

We appreciate the reviewer's insightful comments, which have helped improve the quality and clarity of our manuscript. We hope that the revised version meets the expectations of the editorial team and reviewers.

Sincerely,
Irene Carlino

On behalf of all co-authors

Round 2

Reviewer 2 Report

Comments and Suggestions for Authors

Accept in present form.